# Health Determinants Associated with the Mediterranean Diet: A Cross-Sectional Study

**DOI:** 10.3390/nu14194110

**Published:** 2022-10-03

**Authors:** Nadia San Onofre Bernat, Joan Quiles i Izquierdo, Eva María Trescastro-López

**Affiliations:** 1Department of Community Nursing, Preventive Medicine and Public Health and History of Science, University of Alicante, 03690 Alicante, Spain; 2General Directorate of Public Health and Addictions, Department of Universal Health and Public Health, Generalitat Valenciana, Biomedical Research Consortium in Epidemiology and Public Health Network, 46021 Valencia, Spain; 3Grupo Balmis de Investigación en Historia de la Ciencia, Cuidados en Salud y Alimentación, 03690 Alicante, Spain

**Keywords:** Mediterranean diet, lifestyles, health, chronic noncommunicable diseases, obesity

## Abstract

Introduction: The Mediterranean diet (MD) has been shown to be a good tool for the prevention of obesity and other chronic noncommunicable diseases (NCDs) and to have a low environmental impact. The aim of the present study was to evaluate the relationship between declared morbidity, lifestyles and other sociodemographic factors with high adherence to the MD (AMD) in an adult population in southeastern Spain. Material and Methods: We conducted a cross-sectional study of a sample (*n* = 2728) representative of a non-institutionalized population ≥16 years. The data corresponded to the 2010-11 Nutrition Survey of the Valencian Community. The AMD was assessed using the Mediterranean Diet Adherence Screener questionnaire. The association of variables and high AMD was assessed by univariate and multivariate logistic regression determining crude and adjusted odds ratios. Results: Multivariate analysis showed that age 45 years or older, living with a partner, eating between meals, and not smoking were associated with high AMD. The age groups 45–64 years and 65 years or older showed the strongest association with high AMD in both sexes. Conclusion: The investigation showed a generational loss of AMD. People older than 45 years and living in company are more likely to adhere to DM, the risk group being young people living alone and smokers.

## 1. Introduction

In recent years, the Spanish population has undergone significant changes in lifestyles and, in particular, in diet, which can have a negative influence on people’s state of health [1]. These changes have resulted in an increase in the rates of overweight, obesity and chronic noncommunicable diseases (NCDs), which have become an important challenge for public health as they reduce the quality of life of the people who suffer from them, as well as that of their families, and lead to an increase in healthcare expenditure [1]. On the other hand, the obesogenic environments in which the current population live have favored the development of obesity by stimulating unhealthy habits or behaviors [2,3].

For these reasons, to reduce the risk factors associated with NCDs, such as overweight and obesity, emphasis should be placed on the promotion of healthy diets, physical activity, smoking cessation, protection against alcohol consumption, as well as support for effective interventions aimed at changing environments so that the healthiest options are also the easiest [4,5]. In this contextual framework, the Mediterranean diet (MD) is presented as a healthy dietary model capable of guaranteeing adequate nutrient intake, preventing and reducing complications associated with NCDs, regulating body weight, and having a low environmental impact [6,7,8,9]. At present, numerous studies suggest that adherence to the Mediterranean Diet (AMD) plays an important role in the primary and secondary prevention of cardiovascular diseases (CVD), in addition to improving health in people with various pathologies [10]. Although it has been shown that AMD can vary according to socioeconomic characteristics of the population, as well as other determinants of health [11,12,13,14]; it is of interest to know the factors that may affect the level of adherence to this dietary pattern in a population in order to be able to carry out equitable actions that facilitate high AMD and better health for all.

For these reasons, the study of adherence to the Mediterranean dietary pattern may be relevant for public health, both in the promotion of healthy lifestyles and in the prevention of chronic diseases and obesity.

The aim of the present study was to evaluate the relationship between declared morbidity, lifestyles and other sociodemographic factors with high AMD in a sample of the adult population of southeastern Spain.

## 2. Materials and Methods

Description of the study: We conducted a cross-sectional population-based study of a representative sample of the adult population (16 years of age or older). The population and intake data used came from the Nutrition Survey of the Valencian Community 2010–2011 (ENCV2010-11) [15], which collects epidemiological data on the non-institutionalized population of southeastern Spain. In the ENCV2010-11, a random cluster sampling was performed, proportional to provincial demarcation and stratified by age and sex groups. The Population Information System of the Conselleria de Sanidad Universal y Salud Pública (Regional Ministry of Universal Health and Public Health) was used as the sampling frame, and the universe was the population residing in southeastern Spain over 6 years of age. The ENCV2010-11 worked with 4500 people, and the final participation rate was 68.9% (3102 individuals). The aim of the ENCV2010-11 was to find out what the eating habits of the children and adult population of southeastern Spain were and how they were related to their state of health.

For the present study, we selected all records corresponding to persons between 16 and 95 years of age who had completed the main questionnaire and the food consumption frequency questionnaire (CFQ) of the ENCV2010-11. The sample used for the analyses consisted of 2728 records corresponding to 1311 men and 1417 women.

Assessment of the AMD: To assess the AMD, the data available in the CFQ (of 142 items) of the origin survey (ENCV2010-11) were adapted to the Mediterranean Diet Adherence Screener (MEDAS) questionnaire [16,17] by converting the food consumption frequencies declared in the CFQ, expressing the results as daily servings. The conversion factors used for each category were standardized by consumption of servings per day, using the following values: less than 1 time per month (0.02 servings/day); less than 1 time per week (0.07 servings/day); 1 time per week (0.14 servings/day); 3 times per week (0.43 servings/day); between 4 and 6 times per week (0.79 servings/day); 1 time per day (1 serving/day).

In order to be able to answer all the questions in the MEDAS questionnaire, some adaptations had to be made, as was the case with the item “sofrito”, which was not included in the CFQ of the ENCV2010-1. For the assessment of the item “traditional tomato sauce” (sofrito) that appears in the MEDAS questionnaire, we proceeded to adapt the variables of the CFQ used, making use of the data on the intake of onion, spring onion, garlic and fried tomato sauce reflected in the CFQ.

Each item of the MEDAS questionnaire was assigned a value of 0 if the condition was not met or a value of 1 if it was met. The criteria followed to assign a value of 1 to each item were as follows: using olive oil as the main fat; eating at least 4 tablespoons (≥60 g) of olive oil throughout the day; eat at least 2 servings (≥300 g) of vegetables per day; eat at least 3 servings (≥360 g) of fruit per day; eat less than 1 serving (<125 g) of red meat and hamburger per day; eat less than 1 serving (<12.5 g) of butter per day; drink less than 1 serving (<250 mL) of sweetened or carbonated beverages per day; drink 7 servings (700 mL) of wine per week or more; eat at least 3 servings (≥240 g raw or 600 g if cooked) of pulses per week; eat at least 3 servings (≥450 g) of fish per week; eat no more than 3 servings (≤150 g) per week of industrial bakery products; eat at least 3 servings (≥90 g) of nuts per week; preferably consuming more white meat than red meat; eating pasta, vegetables, rice or other dishes with soffit at least 2 times per week. Cases that did not meet these conditions were assigned a value of 0.

To estimate the degree of AMD, a sum of the score obtained in each question was made. The possible range of scores ranged from 0 to 14 points, being 0 when no condition defined in the Mediterranean dietary pattern was met and 14 when all conditions were met (14). This allowed the population to be classified into three groups according to the degree of AMD. Scores between 0–5 formed the “low adherence” group, between 6–9 “medium adherence”, and scores between 10 and 14 (both included) made up the group of “high adherence” to DM [17].

For the analyses on the association between AMD and declared morbidity, lifestyles and sociodemographic factors, the population was classified into two groups according to the level of AMD (scores of 0 to 9 formed the “low-medium AMD” group, and scores of 10 to 14 formed the “high AMD” group), which allowed us to perform binomial logistic regressions.

Study variables: The variables and categories used for the analyses were as follows: sex (male or female); age groups (16–24 years, 25–44 years, 45–64 years, ≥65 years); country of birth (Spain or others); educational level (no education, primary education, secondary education or higher education); employment status (with full-time or part-time paid work or without paid work which in turn included unemployed persons, students, retired, permanently disabled, engaged in housework and other kinds of economic activity); living with a partner (yes or no); perception of health (good health, which in turn included the responses very good, good and acceptable and bad health which included bad and very bad); leisure-time physical activity (little or none or several times a week) [18]; sedentary behavior (<2 h/day or ≥2 h/day); hours of screen time per day of television, computer and Internet (>2 h/day or ≤2 h/day) [19]; smoking (yes, no or ex-smokers); snacking between hours (yes or no), alcohol consumption (no consumption–low-risk consumption, risk consumption or harmful consumption) [20], arterial hypertension (AHT) (yes or no), myocardial infarction (yes or no), neurological diseases (yes, no) hypercholesterolemia (yes or no), cancer (yes or no), diabetes (yes or no), body mass index (BMI) (underweight, normal weight, overweight or obesity) [21].

Alcohol consumption (in grams) was calculated by converting the frequency of consumption of wine, cava, beer, spirits and distillates (whiskey, gin, vodka) declared in the CFQ, to daily servings, expressing the results as grams of alcohol per day based on the following formula: grams of alcohol = [(alcohol content × volume (cl))/100] × 8. The reference alcohol content used in the conversions was 12% for wine and champagne, 5% for beer, 25% for spirits and 40% for distillates [17].The total grams of alcohol consumed per day were then summed and categorized into 3 groups: no consumption–low risk consumption (<20 g/day in women, <40 g/day in men), risk consumption (20–40 g/day in women, 40–60 g/day in men) and harmful consumption (>40 g/day in women, >60 g/day in men).

Statistical analysis: IBM^®^ SPSS^®^ Statistics version 27 software (IBM Corp., Armonk, New York, NY, USA) as used for statistical analysis. Quantitative variables were described by means of their mean, standard deviation and medians. The Student t test or analysis of variance (ANOVA) was used to establish differences according to sex and age group, depending on the number of categories considered. For qualitative variables, their frequencies were estimated, and the Chi-square test (X^2^) was used for comparison. A *p*-value of less than 0.05 was established as the level of statistical significance. 

The possible association, for the total sample and stratified by sex, between the dependent variable (AMD) and the rest of the variables was explored with the aim of making a first approximation to the estimation of the measure of association, classifying the variables according to the value of the odds ratio (OR) as a measure of the strength of association, their corresponding 95% confidence intervals and the statistical significance in the contrast to the hypothesis X^2^. 

To evaluate the possible association, a univariate binary logistic regression analysis was performed to explore the distribution. We worked with dichotomous categorical variables except for age groups, educational level, smoking, alcohol consumption and BMI, which were polytomous and were included in the analysis as dummy variables.

A multivariate analysis was performed for the total sample and by sex by means of logistic regression. The variables included in the analysis were those that could cause confusion according to current scientific evidence and those with *p* < 0.2 in the univariate analysis. This procedure made it possible to obtain adjusted measures of association (OR) with 95% confidence intervals and the contrast of hypotheses by means of the Wald statistical test.

## 3. Results

### 3.1. Dscriptive Analysis

Table 1 and Table 2 show the distribution of the sample according to the health determinants studied based on the level of AMD.

The study showed that 18.2% of the population was classified as having high AMD, and that the AMD level of men and women was similar. No significant differences (*p* > 0.05) were found between sexes, but the level of AMD did differ between age groups. 

When analyzing the proportion of the population classified as high AMD (Table 1) according to age group, it was observed that the population aged 16–24 years had the least number of respondents classified in this category. Only 6.6% (*n* = 23) of the total (*n* = 352) respondents in the 16–24 age group had high AMD. The percentages of those classified in the high AMD group increased in parallel with the age group so that 12% (*n* = 113) of the total (*n* = 940) population aged 25–44 years had high AMD; 24.6% (*n* = 208) of the total (*n* = 846) population aged 45–64 years had high AMD; 26% (*n* = 153) of the total (*n* = 589) population aged ≥65 years had high AMD. 

Regarding the relationship between BMI and AMD, Table 3 shows the distribution of the sample according to the declared anthropometry as a function of the AMD level. 

### 3.2. Univariate Analysis

Univariate analysis showed the association between sociodemographic factors, lifestyles and health–disease states with high AMD (Table 4).

Among the variables referring to the sociodemographic situation of the sample, it was observed that there was a positive association between age group and high AMD, and the strength of the association increased as the age of the study group increased. The group aged ≥65 years obtained an OR = 5.04 and 95% CI = 3.18–7.98. The variable country of birth showed significant results in the male group. It was observed that those who were not born in Spain had a negative association with high AMD (OR = 0.44; 95% CI = 0.25–0.76). Educational level also showed an association with high AMD. Having secondary and higher education levels was negatively associated with high AMD in the whole population (OR = 0.60; 95% CI = 0.42–0.86). Employment status was a variable that remained associated with high AMD. The condition of being without paid work was negatively associated with high AMD for the whole population and for men (OR = 0.70; 95% CI = 0.53–0.94). The type of cohabitation showed statistical significance in the analysis of the association with AMD. Not living with a partner was negatively associated with high AMD in all cases: in the general population (OR = 0.48; 95% CI = 0.38–0.60), in the men’s group (OR = 0.39; 95% CI = 0.27–0.55) and in the women’s group (OR = 0.56; 95% CI = 0.42–0.76).

Lifestyle variables were also associated with high AMD. Smoking was significantly associated with high AMD, with a negative association between smoking and high AMD in the whole population (OR = 0.76; 95% CI = 0.60–0.96). Eating between meals was also negatively associated with high AMD in the whole population (OR = 0.63; 95% CI = 0.51–0.78), in men (OR = 0.62; 95% CI = 0.46–0.85) and in women (OR = 0.64; 95% CI = 0.48–0.86). Risky alcohol consumption was positively associated in all cases with high AMD (OR = 2.17; 95% CI = 1.37–3.43) but not with harmful consumption (OR = 1.20; 95% CI = 0.52–2.78). The practice of leisure-time physical activity and sedentary behavior did not obtain statistically significant results in the study of the association with high AMD.

Health–disease status was also associated with high AMD. Having HT was positively associated with high AMD in the general population (OR = 1.81; 95% CI = 1.44–2.26), in men (OR = 1.93; 95% CI = 1.40–2.67), and in women (OR = 1.70; 95% CI = 1.24–2.32). Hypercholesterolemia was positively associated with high AMD in the general population (OR = 1.88; 95% CI = 1.49–2.39), in men (OR = 2.02; 95% CI = 1.43–2.87), and in women (OR = 1.77; 95% CI = 1.28–2.44). Having diabetes was also positively associated with high AMD in men (OR = 1.70; 95% CI = 1.07–2.68). Having suffered a myocardial infarction was positively associated with high AMD in the general population (OR = 2.25; 95% CI = 1.34–3.77) and in men (OR = 2.69; 95% CI = 1.48–4.88). Suffering from neurological diseases was negatively associated with high AMD in women (OR = 0.26; 95% CI = 0.08–0.85). The perception of health status did not show statistical significance in the study of the association with high AMD. 

### 3.3. Multivariate Analysis

Multivariate analysis (Table 5, Table 6 and Table 7) showed the association between the study variables and high AMD by means of adjusted ORs and their 95% CIs.

The variables included in the model for the general population are shown in Table 5, of which age group, cohabitation with a partner and snacking between meals were statistically significant (*p* < 0.05). The results of the multivariate analysis for men are shown in Table 6. In the men’s group, the variables of age group, cohabitation and smoking were significantly associated with high AMD. In the women’s group, the variables of age group and cohabitation were statistically significant, as shown in Table 7.

## 4. Discussion

The present study evaluated the risk factors associated with adherence to the Mediterranean dietary pattern in the adult population of southeastern Spain. The results provide a clearer view of the sociodemographic factors, lifestyles, and health–disease states associated with high AMD in order to propose interventions to promote well-being at different stages of life and reduce morbidity and premature mortality due to NCDs, which is one of the Sustainable Development Goals [22].

The results obtained indicated that less than one-fifth of the population followed traditional, healthy, and sustainable MD, which is in agreement with the results of other studies performed in the general Spanish population [23]. Apart from the low proportion of the population classified at a high level of AMD, the results of the present study also showed that the younger population had the highest proportion of those classified in the low AMD group and the lowest proportion of those classified in the high AMD group. The results of the present work show how the older age groups, which are those that experienced the Spanish epidemiological–nutritional transition in the 1960s and 1970s [24], have a higher AMD. The progressive incorporation of dietary patterns typical of Western countries, which fundamentally affect younger groups, distances the young Spanish population from the traditional MD. This study highlights the generational loss of the MD and the shift away from the high AMD that is occurring in younger age groups. It would therefore be necessary to direct MD promotion and adherence actions to the groups with the lowest adherence as a public health strategy and for the prevention of overweight and obesity.

In recent years, the prevalence of overweight and obesity along with NCDs has steadily increased worldwide and the *European Regional Obesity Report 2022* [25] points out that Spain is one of the European countries with the highest prevalence of childhood obesity. The World Health Organization (WHO) [26] explains that unhealthy dietary patterns and sedentary lifestyles could be some of the causes of weight gain in the population. Actions should therefore be promoted in this regard. There is an urgent need to promote dietary patterns that guarantee the well-being of people and planetary health (as is the case with the Mediterranean diet). The results of this research show that the young population of southeastern Spain is the segment of the population that most departs from the Mediterranean dietary pattern, and this departure can lead to health problems (as explained by the WHO). The predominance of Westernized dietary patterns in the general population and more specifically among the young population is no coincidence. Rather, it is a combination of factors that have resulted in the deterioration of the quality of the population’s diet. In recent decades, as a result of changes in the global food supply [27], high and middle income countries have seen food environments saturated with unhealthy, highly accessible, relatively cheap and highly promoted foods, with easy access to foods high in sodium, saturated fats and/or added sugars [28]. These obesogenic environments encourage overconsumption of energy-dense, nutrient-poor foods. Given this situation, it is recommended at the global level to favour regulations to restrict exposure of children in terms of marketing unhealthy foods, better labelling of products and policies that encourage the consumption of healthy and sustainable foods [29].

The sociodemographic and economic changes that occurred from the mid-20th century to the present also favored the change in the pattern of food consumption in Mediterranean areas [24], the shift away from the Mediterranean dietary pattern and the increase in NCDs and rates of overweight and obesity [30,31]. The present study detected the association between sociodemographic factors with high AMD. On the one hand, the condition of living in a couple was positively associated with high AMD. Previous research [32] had already detected that people living alone were more likely to have deficient intakes of staple foods such as fruits, vegetables, and fish compared to those living with a partner. On the other hand, it was observed that male smokers had a negative association with high AMD, i.e., they had a higher risk of having a low AMD. This same phenomenon was already observed in other studies in which smokers had a low AMD [33]. As a consequence of the results obtained in the present study, as well as the data available in the scientific literature, it would be of interest to study the level of AMD in individuals with toxic habits (such as the consumption of legal and illegal drugs) in order to explain the observed phenomenon in greater detail, since smoking is a toxic habit that should be adequately controlled through the application of effective actions [34]. Despite the advances that have been made [35], smoking continues to be an important public health problem that causes cancer and respiratory and cardiovascular diseases and is one of the main causes of preventable death [36].

Snacking between meals was also negatively associated with high AMD. In a first estimation of the association, it showed statistical significance for the whole population and in both sexes. After adjusting the analysis in the multivariate model, the variable showed statistical significance in the general population, and it was observed that snacking between meals was associated with lower AMD. Dietary guidelines recommend eating three main meals a day [37] to promote dietary balance. However, snacking between meals implies an alteration of the usual eating schedules, an increase in daily food intake and therefore an alteration of circadian rhythms with possible harmful consequences for health [38]. Apart from the negative impact on health of altered mealtimes, in the present study we observed a shift away from MD that may be a consequence of the nutritional quality of the foods chosen for these snacks. If highly processed foods are introduced during snacking, the quality of the diet will be diminished [39] because they are usually products that induce excessive intake of free sugar [40], low-quality fats and, in short, foods that are nutritionally uninteresting and dispensable in a healthy diet [41]. All this is closely related to body weight, since the recurrent intake of this type of products favors an increase in dietary energy, which can result in excess body weight [42]. In any case, there is a clear need to prioritize fresh and plant-based foods and to promote eating habits that emphasize quality, quantity and meal times, as three fundamental aspects to achieve healthy lifestyles [43,44].

It is noteworthy that variables such as educational level, physical activity, sedentary lifestyle, hypercholesterolemia and BMI, did not obtain statistical significance in the multivariate model in any of the cases since scientific evidence closely relates them to AMD. 

In the present study, in a first estimation of risk, an association was observed between health–disease status and high AMD, in both men and women. The population with chronic health problems showed a positive association with high AMD, although when the analysis was adjusted, the statistical significance of the observed phenomenon disappeared. Some studies explain the phenomenon that the majority of people with these pathologies present a moderate AMD and associate it with sociodemographic factors and health status [45,46,47]. Likewise, other studies indicate that dietary recommendations play an important role in the prevention of certain diseases and that MD should be promoted among the population with cardiovascular pathologies [48]. Therefore, it is not surprising that those who suffer from CVD or related health problems have a higher AMD than those who do not, although it would be of interest to find out when this change in diet occurred. Due to the typology of the study (an official survey of a public health agency and endorsed by the Department of Health and the General Directorate of Public Health, with a scope of 4500 people), we could not add an additional questionnaire to solve the dilemma, but we can explain and justify it with the scientific literature and propose that for future occasions this section be considered. The authors point out the importance of carrying out further research that contemplates people with pathologies in order to provide an answer to the unknown. However, it is to be expected that in the population with CVD, cancer and other chronic NCDs, a change in dietary pattern will occur after their detection and diagnosis, since diet plays a very important role in the prevention of these pathologies, and specifically the MD is referenced in many medical guides as a recommended dietary pattern [49,50,51].

Improving diet is a public health priority that can lead to a significant reduction in CVD morbidity and mortality [52]. It is therefore important for healthcare providers to understand current dietary practice guidelines and to apply evidence-based dietary counseling to persons at high risk for CVD. In addition, there is evidence demonstrating that a change in dietary pattern after the diagnosis of some disease or pathology can be explained by the “health belief model” [53]. This model pays preferential attention to the role of the individual’s perception or belief about his or her vulnerability to a disease that threatens his or her health and the actions he or she can take to prevent that threat and avoid the possible disease. According to this model, the person has to feel threatened by his or her previous or current pattern of behavior (in this case a poor diet) in order to make the change towards healthier actions (in this case, the MD) [54].

In view of the above, in our study sample we observed participants modifying their diet towards a Mediterranean dietary pattern to improve their health when confronted with disease diagnoses. It would be interesting to conduct future cohort studies along these lines that would allow us to confirm these observations. 

### Limitations

The present investigation was able to use data from a regional nutrition and health survey to assess AMD using the validated MEDAS questionnaire, although not all available survey data could be used due to the lack of responses from participants.

Regarding the methodology and instruments used, the choice of the diet quality index is one of the most relevant aspects of the study since it establishes the cut-off points on which the conclusions will be drawn. Although diet quality indices are very useful for assessing adherence to dietary patterns in specific populations [18], there is no universal indicator that can be extrapolated to all populations.

In relation to the study sample, the present study evaluated the quality of the diet in relation to the pattern of MD and the factors associated with its practice in the adult population of southeastern Spain but did not consider the population under 16 years of age.

For future lines of research, it would be of interest to observe whether changes in dietary patterns and other lifestyles occur after the diagnosis of CVD and whether these are associated with the traditional Mediterranean pattern. Moreover, as this is a cross-sectional epidemiological study, the results obtained do not allow us to affirm that those with CVD have always had a high AMD. It would be of interest to analyze this phenomenon by means of different surveys (past and future) to be able to observe this change in the pattern of food consumption and corroborate the hypothesis. It would also be of interest to know how the elements of the food and physical environment favor or hinder high AMD and other healthy lifestyles. It would also be interesting to widen the study area to have a clearer and more detailed view of AMD, making use of the different nutrition and health surveys available.

## 5. Conclusions

The present study evaluated the relationship between declared morbidity, lifestyles, and other sociodemographic factors associated with high AMD in a sample of the adult population of southeastern Spain. Age, type of cohabitation, and smoking habit were found to be related to the level of AMD. 

Evidence suggests that MD has salutogenic effects in healthy populations, preventing overweight and obesity and other NCDs, and that it has a lower environmental footprint than the current dietary pattern of the Spanish and Western population [55]. Therefore, it is a priority for public health to promote MD adherence. Likewise, public health actions and policies should focus their efforts on promoting MD in younger groups living alone and on maintaining adherence in older groups. This association should be analyzed in depth, and priority should be given to interventions aimed at the most vulnerable groups in order to facilitate the adoption of healthy lifestyles among the entire population. Likewise, health interventions with a multifactorial approach, including diet and smoking, can be positive in improving lifestyles.

## Figures and Tables

**Table 1 nutrients-14-04110-t001:** Distribution of the sample according to the determinants of health based on the level of adherence to the Mediterranean diet.

	General			Men			Women		
Variables	Adherence to the Mediterranean Diet		Total N (%)	Adherence to the Mediterranean Diet		Total N (%)	Adherence to the Mediterranean Diet		Total N (%)
	≤9 Points N (%)	10–14 Points N (%)		≤9 Points N (%)	10–14 Points N (%)		≤9 Points N (%)	10–14 Points N (%)	
**Age groups ^abc^**									
16–24 years old	330 (14.8)	23 (4.63)	353 (12.9)	161 (14.98)	9 (3.81)	170 (13)	169 (14.6)	14 (5.36)	183 (12.9)
25–44 years old	827 (37.1)	113 (22.7)	940 (34.5)	413 (38.42)	52 (22)	465 (35.5)	414 (35.8)	61 (23.7)	475 (33.5)
45–64 years old	638 (28.6)	208 (41.9)	846 (31)	312 (29.02)	89 (37.7)	401 (30.6)	326 (28.2)	119 (45.6)	445 (31.4)
≥65 years	436 (19.5)	153 (30.8)	589 (21.6)	189 (17.58)	86 (36.4)	275 (21)	247 (21.4)	67 (25.7)	314 (22.2)
Total	2231 (100)	497 (100)	2728 (100)	1075 (100)	236 (100)	1311 (100)	1156 (100)	261 (100)	1417 (100)
**Country of birth ^b^**									
Different from Spain	290 (13.1)	42 (8.47)	332 (12.2)	144 (13.4)	15 (6.38)	159 (12.2)	146 (12.7)	27 (10.3)	173 (12.3)
Spain	1932 (87)	454 (91.5)	2386 (87.8)	929 (86.6)	220 (93.6)	1149 (87.8)	1003 (87.3)	234 (89.7)	1237 (87.7)
Total	2222 (100)	49 (100)	2718 (100)	1073 (100)	235 (100)	1308 (100)	1149 (100)	261 (100)	1410 (100)
**Level of education ^ab^**									
No education	203 (9.15)	63 (12.7)	266 (9.80)	72 (6.75)	31 (13.1)	103 (7.90)	131 (11.4)	32 (12.4)	163 (11.6)
Primary education	453 (20.4)	143 (28.9)	596 (22)	210 (19.7)	74 (31.4)	284 (21.8)	243 (21.1)	69 (26.6)	312 (22.1)
Secondary education	1066 (48)	197 (39.8)	1263 (46.5)	552 (51.7)	90 (38.1)	642 (49.3)	514 (44.6)	107 (41.3)	621 (44)
Higher education	497 (22.4)	92 (18.6)	589 (21.7)	233 (21.8)	41 (17.4)	274 (21)	264 (22.9)	51 (19.7)	315 (22)
Total	2219 (100)	495 (100)	2714 (100)	1067 (100)	236 (100)	1303 (100)	1152 (100)	259 (100)	1411 (100)
**Employment status**									
With paid work	900 (40.5)	181 (36.4)	1.081 (39.8)	576 (53.7)	147 (62.3)	585 (44.7)	745 (64.8)	169 (64.8)	496 (35.2)
No paid work	1321 (59.5)	316 (63.6)	1637 (60.2)	496 (46.3)	89 (37.7)	723 (55.3)	404 (35.2)	92 (35.3)	914 (64.8)
Total	2221 (100)	497 (100)	2718 (100)	1072 (100)	236 (100)	1308 (100)	1149 (100)	261 (100)	1410 (100)
**Living together as a couple ^abc^**									
Yes	1256 (59.2)	353 (75.3)	1609 (62.1)	616 (60.7)	180 (80)	796 (64.2)	640 (57.8)	173 (70.9)	813 (60.2)
No	866 (40.8)	116 (24.7)	982 (37.9)	399 (39.3)	45 (20)	444 (35.8)	467 (42.2)	71 (29.1)	538 (39.8)
Total	2122 (100)	469 (100)	2591 (100)	1015 (100)	225 (100)	1240 (100)	1107 (100)	244 (100)	1351 (100)
**Health perception**									
Good condition	2009 (93.1)	438 (90.9)	2447 (92.7)	974 (94.3)	207 (92)	1181 (93.9)	90 (8)	26 (10.1)	1266 (91.6)
Poor condition	149 (6.90)	44 (9.13)	193 (7.31)	59 (5.71)	18 (8)	77 (6.12)	1035(92.00)	231 (89.9)	116 (8.39)
Total	2158 (100)	482 (100)	2640 (100)	1033 (100)	225 (100)	1258 (100)	1125 (100)	257 (100)	1382 (100)
**Leisure time physical activity**									
Little or no physical activity	1934 (87.9)	440 (89.4)	2374 (88.2)	906 (85.6)	24 (10.3)	1115 (86.4)	1028(90)	231 (89.2)	1.259 (89.9)
Physical activity several times a week	266 (12.1)	52(10.6)	318 (11.8)	152 (14.4)	209 (89.7)	176 (13.6)	114 (10)	28 (10.8)	142 (10.1)
Total	2200 (100)	492 (100)	2692 (100)	1058 (100)	233 (100)	1291 (100)	1142 (100)	259 (100)	1401 (100)
**Sedentary behavior**									
<2 h/day	496 (22.8)	88 (18.6)	584 (22.1)	264 (25.3)	44 (19.7)	308 (24.3)	232 (20.5)	44 (17.6)	276 (20)
≥2 h/day	1679 (77.2)	385 (81.4)	2064 (78)	780 (74.7)	179 (80.3)	959 (75.7)	899 (79.5)	206 (82.4)	1105 (80)
Total	2175 (100)	473 (100)	2648 (100)	1044 (100)	223 (100)	1267 (100)	1131 (100)	250 (100)	1381 (100)
**Current smoking habit**									
Yes	677 (30.4)	122 (24.7)	799 (29.3)	383 (35.6)	69 (29.4)	452 (34.5)	294 (25.4)	53 (20.4)	347 (24.5)
No	1099 (49.3)	261 (52.7)	1360 (49.9)	419 (39)	94 (40)	513 (39.2)	680 (58.8)	167 (64.2)	847 (59.8)
Ex-smokers	455 (20.4)	112 (22.6)	567 (20.8)	273 (25.4)	72 (30.6)	345 (26.3)	182 (15.7)	40 (15.4)	222 (15.7)
Total	2231 (100)	495 (100)	2726 (100)	1075 (100)	235 (100)	1310 (100)	1156 (100)	260 (100)	1416 (100)
**Snacking between meals ^abc^**									
Yes	892 (40.7)	149 (30.3)	1041 (38.8)	427 (40.8)	70 (30)	497 (38.8)	465 (40.6)	79 (30.5)	544 (38.8)
No	1300 (59.3)	343 (69.7)	1643 (61.2)	620 (59.2)	163 (70)	783 (61.2)	680 (59.4)	180 (69.5)	860 (61.3)
Total	2192 (100)	492 (100)	2684 (100)	1047 (100)	233 (100)	1280 (100)	1145 (100)	259 (100)	1404 (100)
**Alcohol consumption ^b^**									
No consumption and low-risk consumption	2144 (96.1)	462 (93)	2606 (95.5)	1018 (94.7)	213 (90.2)	1231 (93.9)	1126 (97.4)	249 (95.4)	1375 (97)
Risky consumption	60 (2.69)	28 (5.63)	88 (3.2)	37 (3.4)	17 (7.20)	54 (4.12)	23 (2)	11 (4.22)	34 (2.40)
Harmful consumption	27 (1.21)	7 (1.41)	34 (1.3)	20 (1.9)	6 (2.54)	26 (2)	7 (0.61)	1 (0.38)	8 (0.56)
Total	2231 (100)	497 (100)	2728 (100)	1075 (100)	236 (100)	1311 (100)	1156 (100)	261 (100)	1417 (100)

^a^: *p* value < 0.05 in general; ^b^: *p* value < 0.05 in men; ^c^: *p* value < 0.05 in women.

**Table 2 nutrients-14-04110-t002:** Distribution of the sample according to the health status based on the level of adherence to the Mediterranean diet.

	General			Men			Women		
Variables	Adherence to the Mediterranean Diet		Total N (%)	Adherence to the Mediterranean diet		Total N (%)	Adherence to the Mediterranean Diet		Total N (%)
	≤9 Points N (%)	10–14 Points N (%)		≤9 Points N (%)	10–14 Points N (%)		≤9 Points N (%)	10–14 Points N (%)	
**Arterial Hypertension**									
Yes	397 (17.8)	140 (28.2)	537 (19.7)	189 (17.6)	69 (29.2)	258 (19.7)	208 (18.1)	71 (27.2)	279 (19.7)
No	1828 (82.2)	357 (71.8)	2185 (80.3)	884 (82.4)	167 (70.8)	1051 (80.3)	944 (81.9)	190 (72.8)	1134 (80.3)
Total	2225 (100)	497 (100)	2.722 (100)	1073 (100)	236 (100)	1309 (100)	1152 (100)	261 (100)	1413 (100)
**Infarction**									
Yes	45 (2.02)	22 (4.43)	67 (2.46)	32 (3)	18 (7.63)	50 (3.82)	13 (1.13)	4 (1.53)	17 (1.20)
No	2181 (98)	475 (95.6)	2656 (97.5)	1042 (97)	218 (92.4)	1260 (96.2)	1139 (98.9)	257 (98.5)	1396 (98.8)
Total	2226 (100)	497 (100)	2723 (100)	1074 (100)	236 (100)	1310 (100)	1152 (100)	261 (100)	1413 (100)
**Diabetes**									
Yes	162 (7.27)	46 (9.26)	208 (7.64)	79 (7.36)	28 (11.9)	107 (8.17)	83 (7.20)	18 (6.90)	101 (7.14)
No	2065 (92.7)	451 (90.7)	2516 (92.4)	995 (92.6)	208 (88.1)	1203 (91.8)	1070 (92.8)	243 (93.1)	1313 (92.9)
Total	2227 (100)	497 (100)	2724 (100)	1074 (100)	236 (100)	1310 (100)	1153 (100)	261 (100)	1414 (100)
**Neurological disease**									
Yes	76 (3.41)	12 (2.42)	88 (3.23)	27 (2.52)	9 (3.81)	36 (2.75)	49 (4.25)	3 (1.15)	52 (3.68)
No	2151 (96.6)	484 (97.6)	2635 (96.8)	1046 (97.5)	227 (96.2)	1273 (97.2)	1105 (95.8)	257 (98.8)	1362 (96.32)
Total	2227 (100)	496 (100)	2723 (100)	1073 (100)	236 (100)	1309 (100)	1154 (100)	260 (100)	1414 (100)
**Hypercholesterolemia**									
Yes	322 (14.5)	120 (24.1)	442 (16.2)	143 (13.3)	56 (23.7)	199 (15.2)	179 (15.5)	64 (24.5)	243 (17.2)
No	1904 (85.5)	377 (75.9)	2281 (83.8)	930 (86.7)	180 (76.3)	1110 (84.8)	974 (84.5)	197 (75.5)	1171 (82.8)
Total	2226 (100)	497 (100)	2723 (100)	1073 (100)	236 (100)	1309 (10)	1153 (100)	261 (100)	1414 (100)
**Cancer**									
Yes	52 (2.34)	15 (3.02)	67 (2.46)	17 (1.58)	6 (2.54)	23 (1.76)	35 (3.04)	9 (3.45)	44 (3.11)
No	2173 (97.7)	482 (97)	2655 (97.5)	1056 (98.4)	230 (97.5)	1286 (98.2)	1117 (97)	252 (96.6)	1369 (96.9)
Total	2225 (100)	497 (100)	2722 (100)	1073 (100)	236 (100)	1309 (100)	1152 (100)	261 (100)	1413 (100)
**Body mass index (kg/m^2^)**									
Underweight (<18.5 kg/m^2^)	58 (2.81)	13 (2.86)	71 (2.82)	11 (1.08)	0 (0,00)	11 (0.89)	47 (4.50)	13 (5.60)	60 (4.70)
Normal weight (18.5–24.99 kg/m^2^)	973 (47.2)	181 (39.8)	1154 (45.8)	408 (40.1)	78 (35)	486 (39.2)	565 (54.1)	103 (44.4)	668 (52.4)
Overweight (25–30 kg/m^2^)	710 (34.4)	381 (83.7)	1091 (43.3)	436 (42.8)	107 (48)	543 (43.8)	274 (26.3)	76 (32.8)	350 (27.4)
Obesity (>30 kg/m^2^)	321 (15.6)	78 (17.1)	399 (15.8)	163 (16)	38 (17)	201 (16.2)	158 (11.1)	40 (17.2)	198 (15.5)
Total	2063 (100)	455 (100)	2518 (100)	1018 (100)	223 (100)	1241 (100)	1044(100)	232(100)	1276 (100)

**Table 3 nutrients-14-04110-t003:** Adherence to the Mediterranean diet in an adult population of southeastern Spain according to body mass index.

MEDAS * Score	N	Median BMI ^†^	Standard Deviation	Standard Error	95% Confidence Interval for the Mean		*p*-Value
					Lower Limit	Upper Limit	
2	5	22.21	4.368	1.95	16.79	27.64	0.002
3	18	24.20	4.98	1.17	21.72	26.68	
4	60	23.57	4.38	0.56	22.44	24.70	
5	183	24.65	5.27	0.39	23.89	25.42	
6	312	25.23	4.94	0.28	24.69	25.79	
7	484	26.08	9.14	0.42	25.26	26.89	
8	565	26.19	5.19	0.22	25.76	26.62	
9	436	26.06	4.79	0.23	25.61	26.51	
10	300	26.29	4.99	0.29	25.73	26.86	
11	124	26.48	3.53	0.32	25.85	27.11	
12	27	24.93	3.30	0.64	23.62	26.24	
13	5	25.77	1.52	0.68	23.89	27.65	
Total	2519	25.84	5.98	0.12	25.61	26.08	

* MEDAS: Mediterranean Diet Adherence Screener. ^†^ BMI: Body Mass Index

**Table 4 nutrients-14-04110-t004:** Univariate analysis of risk factors related to high adherence to the Mediterranean diet in the adult population of southeastern Spain.

Univariate	General		Men		Women	
Variables	*p*-Value	Odds Ratio (95% Confidence Interval)	*p*-Value	Odds Ratio (95% Confidence Interval)	*p*-Value	Odds Ratio (95% Confidence Interval)
Sex						
Men		1 (ref.)				
Women	0.78	1.03 (0.85–1.25)				
Age groups						
16–24 years old		1 (ref.)		1 (ref.)		1 (ref.)
25–44 years old	0.01	1.96 (1.23–3.13)	<0.01	2.25 (1.09–4.68)	<0.01	1.78 (0.97–3.27)
45–64 years old	<0.01	4.68 (2.98–7.34)	<0.01	5.10 (2.51–10.39)	<0.01	4.41 (2.46–790)
≥65 years	<0.01	5.04 (3.18–7.98)	<0.01	8.14 (3.97–16.69)	0.13	3.27 (1.78–6.02)
Country of birth						
Different from Spain	0.57	0.95 (0.78–1.15)	<0.01	0.44 (0.25–0.76)	0.29	1.26 (0.82–1.95)
Spain		1 (ref.)		1 (ref.)		1 (ref.)
Level of education						
No education		1 (ref.)		1 (ref.)	0.21	1 (ref.)
Primary education	0.92	1.02 (0.72–1.43)	<0.01	0.82 (0.50–1.35)	0.05	1.16 (0.73–1.86)
Secondary education	<0.01	0.60 (0.43–82)	<0.01	0.38 (0.24–0.61)	0.33	0.85 (0.55–1.33)
Higher education	0.01	0.60 (0.42–0.86)	<0.01	0.41 (0.24–0.70)	0.26	0.79 (0.49–1.29)
Employment status						
With paid work		1 (ref.)		1 (ref.)		1 (ref.)
No paid work	0.09	0.84 (0.69–1.03)	0.02	0.70 (0.53–0.94)	0.98	1.00 (0.76–1.33)
Living together as a couple						
Yes		1 (ref.)		1 (ref.)		1 (ref.)
No	<0.01	0.48 (0.38–0.60)	<0.01	0.39 (0.27–0.55)	<0.01	0.56 (0.42–0.76)
Health perception						
Good condition		1 (ref.)		1 (ref.)		1 (ref.)
Poor condition	0.09	1.35 (0.95–1.93)	0.20	1.44 (0.83–2.49)	0.27	1.29 (0.82–2.05)
Leisure time physical activity						
Little or no physical activity	0.34	1.16 (0.85–1.59)	0.10	1.46 (0.93–2.31)	0.69	0.92 (0.59–1.42)
Physical activity several times a week		1 (ref.)		1 (ref.)		1 (ref.)
Sedentary behavior						
<2 h/day		1 (ref.)		1 (ref.)		1 (ref.)
≥2 h/day	0.05	1.29 (1.00–1.66)	0.08	1.38 (0.96–1.97)	0.30	1.21 (0.85–1.73)
Current smoking habit					0.19	
Yes	0.01	0.76 (0.60–0.96)	0.07	1.33 (0.98–1.81)	0.09	1.73 (0.52–1.03)
No		1 (ref.)		1 (ref.)		1 (ref.)
Ex-smokers	0.04	1.04 (0.81–1.33)	0.10	0.80 (0.57–1.13)	0.89	0.90 (0.61–1.31)
Snacking between meals						
Yes	<0.01	0.63 (0.51–0.78)	<0.01	0.62 (0.46–0.85)	<0.01	0.64 (0.48–0.86)
No		1 (ref.)		1 (ref.)		1 (ref.)
Alcohol consumption						
No consumption–low-risk consumption		1 (ref.)		1 (ref.)		1 (ref.)
Risky consumption	<0.01	2.17(1.37–3.43)	0.01	2.20 (1.21–3.97)	0.03	2.16 (1.04–4.50)
Harmful consumption	0.72	1.20 (0.52–2.78)	0.50	1.43 (0.57–3.61)	0.67	0.65 (0.08–5.27)
Arterial hypertension (AHT)						
Yes	<0.01	1.81 (1.44–2.26)	<0.01	1.93 (1.40–2.67)	<0.01	1.70 (1.24–2.32)
No		1 (ref.)		1 (ref.)		1 (ref.)
Hypercholesterolemia						
Yes	<0.01	1.88 (1.49–2.39)	<0.01	2.02 (1.43–2.87)	<0.01	1.77 (1.28–2.44)
No		1 (ref.)		1 (ref.)		1 (ref.)
Diabetes						
Yes	0.13	1.30 (0.92–1.83)	0.02	1.70 (1.07–2.68)	0.86	0.96 (0.56–1.62)
No		1 (ref.)		1 (ref.)		1 (ref.)
Myocardial infarction						
Yes	<0.01	2.25 (1.34–3.77)	<0.01	2.69 (1.48–4.88)	0.59	1.36 (0.44–4.22)
No		1 (ref.)		1 (ref.)		1 (ref.)
Neurological diseases						
Yes	0.26	0.70 (0.38–1.30)	0.27	1.54 (0.71–3.31)	0.03	0.26 (0.08–0.85)
No		1 (ref.)		1 (ref.)		1 (ref.)
Cancer						
Yes	0.38	1.30 (0.73–2.33)	0.32	1.62 (0.63–4.16)	0.73	1.14 (0.54–2.40)
No		1 (ref.)		1 (ref.)		1 (ref.)
Body mass index (BMI)	0.03		0.49		0.06	
Normal Weight				1 (ref.)		1 (ref.)
Underweight	0.56	1.21 (0.65–2.24)	0.99	1 (0.00-)	0.21	1.52 (0.79–2.90)
Overweight	<0.01	1.39 (1.10–1.74)	0.12	1.28 (0.93–1.77)	0.01	1.52 (1.09–2.12)
Obesity	0.07	1.31 (0.97–1.75)	0.36	1.22 (0.79–1.87)	0.11	1.39 (0.93–2.08)

**Table 5 nutrients-14-04110-t005:** Multivariate analysis of risk factors related to high adherence to the Mediterranean diet in the adult population of southeastern Spain.

General Multivariate Model			
Variables	* _Raw_ RO	_Adjusted_ OR (** 95% CI)	*p*-Value
Age groups			
16–24 years old	1 (ref.)	1 (ref.)	<0.01
25–44 years old	1.96	1.98 (1.10–3.59)	
45–64 years old	4.68	4.23 (2.33–7.70)	
≥65 years	5.04	3.61 (1.91–6.80)	
Employment status			
With paid work	1 (ref.)	1 (ref.)	0.29
No paid work	0.84	0.87 (0.66–1.13)	
Living together as a couple			
Yes	1 (ref.)	1 (ref.)	0.02
No	0.48	0.65 (0.50–0.85)	
Health perception			
Good condition	1 (ref.)		0.87
Poor condition	1.35	0.97 (0.62–1.49)	
Snacking between meals			
No	1 (ref.)	1 (ref.)	0.03
Yes	0.63	0.77 (0.59–0.98)	
Sedentary behavior			
<2 h/day	1 (ref.)	1 (ref.)	0.25
≥2 h/day	1.29	1.18 (0.89–1.58)	
Current smoking habit			
No	1 (ref.)	1 (ref.)	0.24
Yes	0.76	0.83 (0,62–1.11)	
Ex-smokers	1.04	0.81 (0.61–1.07)	
Alcohol consumption			
No consumption–low-risk consumption	1 (ref.)	1 (ref.)	0.42
Risky consumption	2.17	1.44 (0.83–2.49)	
Harmful consumption	1.20	1.12 (0.44–2.85)	
Arterial hypertension (AHT)			
No	1 (ref.)	1 (ref.)	0.29
Yes	1.81	1.18 (0.87–1.59)	
Hypercholesterolemia			
No	1 (ref.)	1 (ref.)	0.06
Yes	1.88	1.32 (0.98–1.76)	
Diabetes			
No	1 (ref.)	1 (ref.)	0.50
Yes	1.30	0.86 (0.56–1.33)	
Myocardial infarction			
No	1 (ref.)	1 (ref.)	0.83
Yes	2.25	1.08 (0.54–2.16)	
Body mass index (BMI)			
Normal Weight	1 (ref.)	1 (ref.)	0.09
Underweight	1.21	2.16 (1.07–4.34)	
Overweight	1.39	1.02 (0.79–1.33)	
Obesity	1.31	0.85 (0.60–1.18)	

* OR = odds ratio, ** IC = confidence interval.

**Table 6 nutrients-14-04110-t006:** Multivariate analysis of risk factors related to high adherence to the Mediterranean diet in adult men in southeastern Spain.

Multivariate Model Men			
Variables	* _Raw_ RO	_Adjusted_ OR (** CI 95%)	*p*-Value
Age groups			
16–24 years old	1 (ref.)	1 (ref.)	<0.01
25–44 years old	2.25	2.97 (1.10–8.06)	
45–64 years old	5.10	6.16 (2.25–16.9)	
≥65 years	8.14	8.16 (2.89–23.1)	
Employment status			
With paid work	1 (ref.)	1 (ref.)	0.19
No paid work	0.70	0.77 (0.52–1.14)	
Living together as a couple			
Yes	1 (ref.)	1 (ref.)	0.01
No	0.39	0.59 (0.40–0.90)	
Snacking between meals			
No	1 (ref.)		0.33
Yes	0.62	0.84 (0.59–1.19)	
Leisure time physical activity			
Physical activity several times a week	1 (ref.)	1 (ref.)	0.85
Little or no physical activity	1.46	0.95 (0.58–1.57)	
Sedentary behavior			
<2 h/day	1 (ref.)	1 (ref.)	0.46
≥2 h/day	1.29	1.16 (0.78–1.73)	
Current smoking habit			
No	1 (ref.)	1 (ref.)	0.03
Yes	1.33	0.60 (0.40–0.89)	
Ex-smokers	0.80	0.71 (0.48–1.04)	
Alcohol consumption			
No consumption–low-risk consumption	1 (ref.)	1 (ref.)	0.24
Risky consumption	2.20	1.72 (0.89–3.32)	
Harmful consumption	1.43	1.35 (0.47–3.90)	
Arterial hypertension (AHT)			
No	1 (ref.)	1 (ref.)	0.95
Yes	1.93	1.04 (0.70–1.53)	
Hypercholesterolemia			
No	1 (ref.)	1 (ref.)	0.08
Yes	2.02	1.41 (0.95–2.10)	
Diabetes			
No	1 (ref.)	1 (ref.)	0.71
Yes	1.70	0.90 (0.54–1.53)	
Myocardial infarction			
No	1 (ref.)	1 (ref.)	0.60
Yes	2.69	1.21 (0.60–2.44)	

* OR = odds ratio, ** IC = confidence interval.

**Table 7 nutrients-14-04110-t007:** Multivariate analysis of risk factors related to high adherence to the Mediterranean diet in adult women in southeastern Spain.

Multivariate Model Women			
Variables	* _Raw_ RO	_Adjusted_ OR (** 95% CI)	*p*-Value
Age groups			
16–24 years old	1 (ref.)	1 (ref.)	<0.01
25–44 years old	1.78	1.55 (0.76–3.13)	
45–64 years old	4.41	3.55 (1.76–7.18)	
≥65 years	3.27	2.04 (0.94–4.45)	
Living together as a couple			
Yes	1 (ref.)	1 (ref.)	0.03
No	0.56	0.68 (0.48–0.97)	
Snacking between meals			
No	1 (ref.)		0.07
Yes	0.64	0.74 (0.53–1.02)	
Alcohol consumption			
No consumption–low-risk consumption	1 (ref.)	1 (ref.)	0.79
Risky consumption	2.16	1.20 (0.48–2.96)	
Harmful consumption	0.65	0.54 (0.06–4.62)	
Arterial hypertension (AHT)			
No	1 (ref.)	1 (ref.)	0.11
Yes	1.70	1.41 (0.93–2.14)	
Hypercholesterolemia			
No	1 (ref.)	1 (ref.)	0.18
Yes	1.77	1.31 (0.88–1.95)	
Neurological diseases			
No	1 (ref.)	1 (ref.)	0.81
Yes	0.26	0.19 (0.05–0.82)	
Body mass index (BMI)			
Normal Weight	1 (ref.)	1 (ref.)	0.09
Underweight	1.52	2.43 (1.20–4.92)	
Overweight	1.52	1.04 (0.72–1.50)	
Obesity	1.39	0.93 (0.59–1.45)	

* OR = odds ratio, ** IC = confidence interval.

## Data Availability

Not applicable.

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
