# Peer review of "Health Determinants Associated with the Mediterranean Diet: A Cross-Sectional Study"

_nutrients, 2022, doi:10.3390/nu14194110_

Round 1
Reviewer 1 Report
The research into the health effects of diet is very important. The Mediterranean diet is considered healthy and balanced, and the manuscript looks at the correlation between adherence to the Mediterranean diet and various factors related to life and health. In several respects, the results were quite surprising and their discussion unconvincing.
Correlation analysis showed that the Mediterranean diet is used by people with health problems. The conclusions indicate that subsequent studies should show when these people started using this diet, whether in response to health problems, or have followed the Mediterranean diet for a long time, and poor health is related to its use; which came first. However, this is too serious a problem to leave readers with such a dilemma, but what if there are indeed any adverse effects from following the Mediterranean diet? Before the article is published, this problem needs to be solved, so additional research needs to be carried out in which respondents will answer how long they have been following the Mediterranean diet and how long they have been suffering from health problems.
Other comments:
Noncommunicable or non communicable - the same spelling should be used throughout the text.
Page 2, lines 92-97 servings should be described in more detail e.g. cup, glass etc.
Tables and Figures are distant from the text that describes them, and it is difficult to track the results by reading the text, maybe Table 1 should be divided into several smaller ones. Some Tables are not numbered and not cited in the text.
Line 223-224 is to be removed.
In Table 1 the sum for General is for example 2,728 for age groups, 2,718 for country of birth etc. Why are the sums not the same?
The research area concerned the Comunitat Valenciana or South-East Spain? line 228 vs. 247.
What makes young people move away from the Mediterranean diet, why do they choose to switch to the Western diet? What factors?
The Supplementary Materials section should be supplemented or deleted, the same applies to Institutional Review Board Statement, Informed Consent Statement and Data Availability Statement
The Author Contributions section should be completed according to the involvement of the individual authors.
Author Response
C.1. Correlation analysis showed that the Mediterranean diet is used by people with health problems. The conclusions indicate that subsequent studies should show when these people started using this diet, whether in response to health problems, or have followed the Mediterranean diet for a long time, and poor health is related to its use; which came first. However, this is too serious a problem to leave readers with such a dilemma, but what if there are indeed any adverse effects from following the Mediterranean diet? Before the article is published, this problem needs to be solved, so additional research needs to be carried out in which respondents will answer how long they have been following the Mediterranean diet and how long they have been suffering from health problems.
A.1.
The Mediterranean diet has beneficial effects on health, as demonstrated in the scientific literature. The greater the adherence to the Mediterranean diet, the lower the risk of mortality, so it is to be expected that the population with a high adherence to the Mediterranean diet will have better health than those who follow a more Western dietary pattern. [i],[ii],[iii],[iv],[v],[vi],[vii],
In this cross-sectional observational epidemiological study, the dietary pattern of people suffering from a disease prior to their illness was not studied. On the other hand, the study reveals that at the time of the study, part of the sample studied suffered from a health problem and that this coincided with a good adherence to the Mediterranean diet.
For this reason, the authors point out the importance of further research that includes people with pathologies, in order to answer the question. However, it is to be expected (and this is pointed out in the present study) that, in the population with cardiovascular diseases, cancer and other chronic non-communicable diseases, a change in the dietary pattern will occur after their detection and diagnosis, since diet plays a very important role in the prevention of these pathologies, and specifically the Mediterranean diet is referred to in many medical guides as a dietary pattern to be recommended. [viii],[ix],[x].
Improving diet is a public health priority that can lead to a significant reduction in CVD morbidity and mortality. It is therefore important that healthcare workers understand current dietary practice guidelines and apply evidence-based dietary advice to people at high risk of CVD [xi].
In addition, there is evidence of a change in dietary pattern after the diagnosis of a disease or pathology, which could also be explained by the "health belief model". This model gives preferential attention to the role of the individual's perception or belief about his or her vulnerability to a disease that threatens his or her health and the actions he or she can take to prevent that threat and avoid the possible disease. According to this model, the person has to feel threatened by their previous or current pattern of behaviour (in this case a poor diet) to make the change towards healthier actions (in this case, the Mediterranean diet) [xii]. For all of the above reasons, we explain the phenomenon observed as that after the diagnosis of an illness, the study sample modified their diet and tended towards a Mediterranean dietary pattern to improve their health condition, although it would be interesting to carry out future studies along these lines to provide an answer to the hypothesis put forward.
Due to the typology of the study (an official survey of a public health government department, endorsed by the Department of Health and the Directorate General of Public Health, with a scope of 4500 people) an additional questionnaire cannot be added to resolve the dilemma raised, but it can be explained and justified with the scientific literature and it is proposed that this section be considered for future occasions.
C.2. Noncommunicable or non communicable - the same spelling should be used throughout the text.
A.2. We have revised all the text so that it always appears as "noncommunicable".
C.3. Page 2, lines 92-97 servings should be described in more detail e.g. cup, glass etc.
A.3. To give more detail, the amount in grams or millilitres of the food or food group has been added.
…eat at least 4 tablespoons (≥ 60 grams) of olive oil throughout the day; eat at least 2 servings (≥ 300 grams) of vegetables per day; eat at least 3 servings (≥ 360 grams) of fruit per day; eat less than 1 serving (<125 grams) of red meat and hamburger per day; eat less than 1 serving (<12.5 grams) of butter per day; drink less than 1 serving (<250 ml) of sweetened or carbonated beverages per day; drink 7 servings ( 700 ml) of wine per week or more; eat at least 3 servings (≥ 240 grams raw or 600 grams if cooked) of pulses per week; eat at least 3 servings (≥ 450 grams) of fish per week; eat no more than 3 servings (≤150 grams) per week of industrial bakery products; eat at least 3 servings (≥ 90 grams) of nuts per week;
C.4. Tables and Figures are distant from the text that describes them, and it is difficult to track the results by reading the text, maybe Table 1 should be divided into several smaller ones.
A.4. The table 1 are modified
C.5. Some Tables are not numbered and not cited in the text.
A.5. The text has been revised and the citation of all the tables and figures has been added.
C.6. ine 223-224 is to be removed.
A.6. Lines 223-224 are deleted.
C.7. In Table 1 the sum for General is for example 2,728 for age groups, 2,718 for country of birth etc. Why are the sums not the same?
A.7. The questionnaire used for this nutrition survey by the General Directorate of Public Health consists of some compulsory items such as those corresponding to the identification of the individual (such as age and sex) and other voluntary items. For the selection of the study sample, the inclusion criteria were chosen to be people aged 16 years or older who had completed the main questionnaire (which is compulsory for all) and the food consumption frequency questionnaire (as explained in the methodology). However, those who did not have information on health status or anthropometric variables, for example, were not excluded from the study. It is for these reasons that the total sample may vary according to the variable studied.
C.8. The research area concerned the Comunitat Valenciana or South-East Spain? line 228 vs 247.
A.8. The terminology used is reviewed and the expression is modified to “South-East Spain” so as not to create confusion for readers.
C.9. What makes young people move away from the Mediterranean diet, why do they choose to switch to the Western diet? What factors?
A.9. A more complete reflection of this section is added in the discussion:
In recent years the prevalence of overweight and obesity along with chronic noncommunicable diseases (NCDs) has steadily increased worldwide and the European Regional Obesity Report 2022[xiii] points out that Spain is one of the European countries with the highest prevalence of childhood obesity. The World Health Organization (WHO) [xiv] explains that unhealthy dietary patterns and sedentary lifestyles could be some of the causes of weight gain in the population. Actions should therefore be promoted in this regard. There is an urgent need to promote dietary patterns that guarantee the well-being of people and planetary health (as is the case of the Mediterranean diet). The results of this research show that the young population of south-eastern Spain is the one that most departs from the Mediterranean dietary pattern, and this departure can lead to health problems (as explained by the WHO). The predominance of Westernised dietary patterns in the general population and more specifically among the young population is no coincidence. Rather, it is a combination of factors that have resulted in the deterioration of the quality of the population's diet. In recent decades, as a result of changes in the global food supply[xv], most high and middle income countries have seen food environments saturated with unhealthy, highly accessible, relatively cheap and highly promoted foods, with easy access to foods high in sodium, saturated fats and/or added sugars[xvi]. These obesogenic environments encourage overconsumption of energy-dense, nutrient-poor foods. Given this situation, it is recommended at the global level to favour regulations to restrict exposure to children in terms of marketing unhealthy foods, better labelling of products and policies that encourage the consumption of healthy and sustainable foods.
C.10. The Supplementary Materials section should be supplemented or deleted, the same applies to Institutional Review Board Statement, Informed Consent Statement and Data Availability Statement. The Author Contributions section should be completed according to the involvement of the individual authors.
A.10. This
[i] Sofi F, Cesari F, Abbate R, Gensini GF, Casini A. Adherence to Mediterranean diet and health status: meta-analysis. BMJ. 2008;337:a1344. Published 2008 Sep 11. doi:10.1136/bmj.a1344
[ii] Sofi F, Abbate R, Gensini GF, Casini A. Accruing evidence on benefits of adherence to the Mediterranean diet on health: an updated systematic review and meta-analysis. Am J Clin Nutr. 2010;92(5):1189-1196. doi:10.3945/ajcn.2010.29673
[iii] Biagi C, Nunzio MD, Bordoni A, Gori D, Lanari M. Effect of Adherence to Mediterranean Diet during Pregnancy on Children's Health: A Systematic Review. Nutrients. 2019;11(5):997. Published 2019 May 1. doi:10.3390/nu11050997
[iv] Papadaki A, Nolen-Doerr E, Mantzoros CS. The Effect of the Mediterranean Diet on Metabolic Health: A Systematic Review and Meta-Analysis of Controlled Trials in Adults. Nutrients. 2020;12(11):3342. Published 2020 Oct 30. doi:10.3390/nu12113342
[v] Dominguez LJ, Di Bella G, Veronese N, Barbagallo M. Impact of Mediterranean Diet on Chronic Non-Communicable Diseases and Longevity. Nutrients. 2021;13(6):2028. Published 2021 Jun 12. doi:10.3390/nu13062028
[vi] Ghosh TS, Rampelli S, Jeffery IB, et al. Mediterranean diet intervention alters the gut microbiome in older people reducing frailty and improving health status: the NU-AGE 1-year dietary intervention across five European countries. Gut. 2020;69(7):1218-1228. doi:10.1136/gutjnl-2019-319654
[vii] Morze J, Danielewicz A, Przybyłowicz K, Zeng H, Hoffmann G, Schwingshackl L. An updated systematic review and meta-analysis on adherence to mediterranean diet and risk of cancer. Eur J Nutr. 2021;60(3):1561-1586. doi:10.1007/s00394-020-02346-6
[viii] Magriplis E, Panagiotakos D, Kyrou I, et al. Presence of Hypertension Is Reduced by Mediterranean Diet Adherence in All Individuals with a More Pronounced Effect in the Obese: The Hellenic National Nutrition and Health Survey (HNNHS). Nutrients. 2020;12(3):853. Published 2020 Mar 23. doi:10.3390/nu12030853
[ix] Prieto-González P, Sánchez-Infante J, Fernández-Galván LM. Association between Adherence to the Mediterranean Diet and Anthropometric and Health Variables in College-Aged Males. Nutrients. 2022;14(17):3471. Published 2022 Aug 24. doi:10.3390/nu14173471
[x] Barrea L, Pugliese G, Laudisio D, Colao A, Savastano S, Muscogiuri G. Mediterranean diet as medical prescription in menopausal women with obesity: a practical guide for nutritionists. Crit Rev Food Sci Nutr. 2021;61(7):1201-1211. doi:10.1080/10408398.2020.1755220
[xi] Pallazola VA, Davis DM, Whelton SP, et al. A Clinician's Guide to Healthy Eating for Cardiovascular Disease Prevention. Mayo Clin Proc Innov Qual Outcomes. 2019;3(3):251-267. Published 2019 Aug 1. doi:10.1016/j.mayocpiqo.2019.05.001
[xii] Green L (editorial). The health belief modeland personal health behavior. Health EducationMonographs 1974;2(4):324-5
[xiii] : WHO European Regional Obesity Report 2022. Copenhagen: WHO Regional Office for Europe; 2022
[xiv] WHO. Obesity and overweight https://www.who.int/es/news-room/fact-sheets/detail/obesity-and-overweight
[xv] White M, Aguirre E, Finegood DT, Holmes C, Sacks G, Smith R. What role should the commercial food system play in promoting health through better diet? BMJ. 2020;368:m545. doi: 10.1136/bmj.m545.
[xvi] Swinburn BA, Sacks G, Hall KD, McPherson K, Finegood DT, Moodie ML et al. The global obesity pandemic: shaped by global drivers and local environments. Lancet. 2011;378(9793):804–14. doi: 10.1016/S0140-6736(11)60813-1.

Reviewer 2 Report
The authors conducted an observational study examining the associations of morbidity, lifestyles, and other sociodemographic factors with adherence to the Mediterranean diet in adults. By analyzing the data of 2728 participants of the 2010-11 Nutrition Survey of the Valencian Community, the authors noted that being older, living with a partner, not snacking between meals, and not smoking were associated with high adherence to the Mediterranean diet. Though cross-sectional, these associations may have implications for future research directions, including intervention trials.
There are some comments.
Comments:
1. Introduction: Certain morbidities, lifestyles, and sociodemographic factors were examined in this study. However, it is unclear why the authors hypothesized the associations between these factors and Mediterranean diet adherence in adults. A better-explained rationale is recommended.
2. Results (Line 165-172 on page 4): The authors described that the "younger population had the highest proportion (23.2%) of those classified in the low adherence group and the lowest proportion (6.56%) of those classified in the high adherence group." However, it is unclear what the "younger population" refers to. Also, the percentages (23.2% and 6.56%) were not demonstrated in Table 1. Likewise, the authors described that "the group ≥ 65 years of age having the highest proportion of classified (26%) in high AMD, and the lowest proportion (4.75%) in low AMD." Again, the percentages (26% and 4.75%) were not shown in Table 1.
3. Results (Table 1): The authors presented participants' characteristics according to the levels of adherence to the Mediterranean diet in Table 1. I would suggest conducting statistical tests for the differences between the two adherence groups and presenting the results.
4. Results (Line 178- 214 on pages 4-5): The authors described the results of the univariate analysis. I would suggest presenting the results in a Table.
5. Results (Figure 1): The authors described the distributions of the Mediterranean diet adherence score, a discrete variable, according to BMI categories. I recommend presenting the distributions of the BMI, as a continuous variable, in different adherence groups. Conducting statistical tests for the differences is also recommended.
6. Conclusion: I would recommend beginning with a summary of the findings in this section.
7. Title: I recommend revising the title, so the study design is clearly indicated. An example would be "----: a cross-sectional study." In addition, it is questionable that this study directly addressed the prevention of chronic non-communicable diseases. Notably, this study was an observational study rather than an intervention study. I would suggest deleting the "in the prevention of chronic noncommunicable diseases."
Author Response
C.1. Introduction: Certain morbidities, lifestyles, and sociodemographic factors were examined in this study. However, it is unclear why the authors hypothesized the associations between these factors and Mediterranean diet adherence in adults. A better-explained rationale is recommended.
A.1. The following text is added to improve understanding:
Numerous studies suggest that adherence to DM (ADM) plays an important role in the primary and secondary prevention of cardiovascular disease (CVD), as well as improving health in people with various conditions (10). Although it has been shown that AMD can vary according to socio-economic characteristics of the population, as well as other determinants of health[1],[2],[3],[4] . It is of interest to know the factors that may affect the level of adherence to this dietary pattern in a population in order to be able to carry out equitable actions that facilitate high AMD and better health for all.
C.2. Results (Line 165-172 on page 4): The authors described that the "younger population had the highest proportion (23.2%) of those classified in the low adherence group and the lowest proportion (6.56%) of those classified in the high adherence group." However, it is unclear what the "younger population" refers to. Also, the percentages (23.2% and 6.56%) were not demonstrated in Table 1. Likewise, the authors described that "the group ≥ 65 years of age having the highest proportion of classified (26%) in high AMD, and the lowest proportion (4.75%) in low AMD." Again, the percentages (26% and 4.75%) were not shown in Table 1.
A.2. The instrument used allows the population to be classified into 3 MDA groups (as explained in the methodology): low MDA, medium MDA and high MDA. In this first classification of the population according to MDA level, 23.2% of the population aged 16-24 years were categorised as low MDA. However, in order not to cause confusion to readers, it can be expressed as follows:
When analysing the proportion of the population classified as high MDA (table 1), according to age group, it was observed that the population aged 16-24 years had the least number of respondents classified in this category. 6.6% (n=23) of the total (n=352) respondents in the 16-24 age group had high MDA. The percentages of those classified in the high MDA group increased in parallel with the age group. So that 12% (n=113) of the total (n=940) population aged 25-44 years had high MDA; 24.6% (n=208) of the total (n=846) population aged 45-64 years had high MDA; and 26% (n=153) of the total (n=589) population aged ≥ 65 years had high MDA.
C.3. The authors presented participants' characteristics according to the levels of adherence to the Mediterranean diet in Table 1. I would suggest conducting statistical tests for the differences between the two adherence groups and presenting the results.
A.4. we add in the table the results of the statistical tests
C.4. Results (Line 178- 214 on pages 4-5): The authors described the results of the
univariate analysis. I would suggest presenting the results in a Table.
A.4. Attached table
C.5. Results (Figure 1): The authors described the distributions of the Mediterranean diet adherence score, a discrete variable, according to BMI categories. I recommend presenting the distributions of the BMI, as a continuous variable, in different adherence groups. Conducting statistical tests for the differences is also recommended.
A.5. - We present the BMI distribution as a continuous variable using logistic regression, although the study we present is more qualitative.
C.6. Conclusion: I would recommend beginning with a summary of the findings in this section.
A.6. In the conclusion section it is added:
The present work assessed the relationship between declared morbidity, lifestyles and other socio-demographic factors with high MDA in a sample of the adult population of south-eastern Spain. It was observed that age, type of cohabitation, and smoking habit are related to the level of MDA.
C.7. Title: I recommend revising the title, so the study design is clearly indicated. An example would be "----: a cross-sectional study." In addition, it is questionable that this study directly addressed the prevention of chronic non-communicable diseases. Notably, this study was an observational study rather than an intervention study. I would suggest deleting the "in the prevention of chronic noncommunicable diseases."
A.7. We modify the title:
“Health determinants associated with the Mediterranean Diet: a cross-sectional study”
[1] Alonso-Cabezas M, Pollán M, Alonso-Ledesma I, Fernández de Larrea-Baz N, Lucas P, Sierra Á, Castelló A, Pino MN, Pérez-Gómez B, Martínez-Cortés M, Lope V, Ruiz-Moreno E. Sociodemographic and Lifestyle Determinants of Adherence to Current Dietary Recommendations and Diet Quality in Middle-Aged Spanish Premenopausal Women. Front Nutr. 2022 Jun 14;9:904330. doi: 10.3389/fnut.2022.904330. PMID: 35774550; PMCID: PMC9237508.
[2] Tong TYN, Imamura F, Monsivais P, Brage S, Griffin SJ, Wareham NJ, Forouhi NG. Dietary cost associated with adherence to the Mediterranean diet, and its variation by socio-economic factors in the UK Fenland Study. Br J Nutr. 2018 Mar;119(6):685-694. doi: 10.1017/S0007114517003993. PMID: 29553031; PMCID: PMC5999016.
[3] Bibiloni, M.d.M.; Gallardo-Alfaro, L.; Gómez, S.F.; Wärnberg, J.; Osés-Recalde, M.; González-Gross, M.; Gusi, N.; Aznar, S.; Marín-Cascales, E.; González-Valeiro, M.A.; Serra-Majem, L.; Terrados, N.; Segu, M.; Lassale, C.; Homs, C.; Benavente-Marín, J.C.; Labayen, I.; Zapico, A.G.; Sánchez-Gómez, J.; Jiménez-Zazo, F.; Alcaraz, P.E.; Sevilla-Sánchez, M.; Herrera-Ramos, E.; Pulgar, S.; Sistac, C.; Schröder, H.; Bouzas, C.; Tur, J.A. Determinants of Adherence to the Mediterranean Diet in Spanish Children and Adolescents: The PASOS Study. Nutrients 2022, 14, 738. https://doi.org/10.3390/nu14040738
[4] Mattavelli, E.; Olmastroni, E.; Bonofiglio, D.; Catapano, A.L.; Baragetti, A.; Magni, P. Adherence to the Mediterranean Diet: Impact of Geographical Location of the Observations. Nutrients 2022, 14, 2040. https://doi.org/10.3390/nu14102040

Round 2
Reviewer 1 Report
The article may be accepted for publication in the current version.